# Building Data Triangulation Capacity for Routine Immunization and Vaccine Preventable Disease Surveillance Programs to Identify Immunization Coverage Inequities

**DOI:** 10.3390/vaccines12060646

**Published:** 2024-06-11

**Authors:** Audrey Rachlin, Oluwasegun Joel Adegoke, Rajendra Bohara, Edson Rwagasore, Hassan Sibomana, Adeline Kabeja, Ines Itanga, Samuel Rwunganira, Blaise Mafende Mario, Nahimana Marie Rosette, Ramatu Usman Obansa, Angela Ukpojo Abah, Olorunsogo Bidemi Adeoye, Ester Sikare, Eugene Lam, Christopher S. Murrill, Angela Montesanti Porter

**Affiliations:** 1Global Immunization Division, Centers for Disease Control and Prevention, Atlanta, GA 30333, USA; 2World Health Organization, Dhaka 1212, Bangladesh; 3Rwanda Biomedical Centre, Ministry of Health, Kigali P.O. Box 7162, Rwanda; 4African Field Epidemiology Network, Kigali, Rwanda; 5Health Information Systems Program (HISP), Kigali, Rwanda; mafendeblaise@hisprwanda.org; 6World Health Organization, Kigali P.O. Box 1324, Rwanda; 7National Stop Transmission of Polio (NSTOP) Program, African Field Epidemiology Network (AFENET), Abuja 900103, Nigeria; 8Division of Global Health Protection, Centers for Disease Control and Prevention, Atlanta, GA 30329, USA

**Keywords:** data visualization, capacity building, immunization, surveillance, health equity

## Abstract

The Expanded Programme on Immunization (EPI) and Vaccine Preventable Disease (VPD) Surveillance (VPDS) programs generate multiple data sources (e.g., routine administrative data, VPD case data, and coverage surveys). However, there are challenges with the use of these siloed data for programmatic decision-making, including poor data accessibility and lack of timely analysis, contributing to missed vaccinations, immunity gaps, and, consequently, VPD outbreaks in populations with limited access to immunization and basic healthcare services. Data triangulation, or the integration of multiple data sources, can be used to improve the availability of key indicators for identifying immunization coverage gaps, under-immunized (UI) and un-immunized (zero-dose (ZD)) children, and for assessing program performance at all levels of the healthcare system. Here, we describe the data triangulation processes, prioritization of indicators, and capacity building efforts in Bangladesh, Nigeria, and Rwanda. We also describe the analyses used to generate meaningful data, key indicators used to identify immunization coverage inequities and performance gaps, and key lessons learned. Triangulation processes and lessons learned may be leveraged by other countries, potentially leading to programmatic changes that promote improved access and utilization of vaccination services through the identification of UI and ZD children.

## 1. Background

The Immunization Agenda 2030 (IA2030), a global immunization strategy endorsed by the World Health Assembly in 2020, envisions a world “where everyone, everywhere, at every age, fully benefits from vaccines” [1]. The strategy emphasizes “data-guided” decision-making as a fundamental component of any successful immunization program, necessary to direct strategies to achieve program targets. Greater data use can lead to better-quality data and ultimately contribute to improved immunization program performance by identifying and targeting those who are eligible for vaccination [2,3]. Improved data quality and use are also critical for measuring progress towards achieving the Sustainable Development Goals (SDGs) and Universal Health Coverage (UHC) targets [2,4,5], and for identifying underserved populations for vaccination to achieve measurable reductions in mortality and morbidity from targeted vaccine preventable diseases (VPDs), as highlighted in the U.S. Centers for Disease Control and Prevention Global Health Equity Strategy 2022–2027 [6].

The COVID-19 pandemic affected immunization programs worldwide and highlighted equity issues in immunization coverage, including outreach services commonly used in many low- and middle-income countries with inadequate access to health facilities [7]. In 2021, the number of infants who did not receive the first dose of the diphtheria-tetanus-pertussis-containing vaccine (DTPcv1) was 37% (18.2 million) higher than in 2019 (13.3 million) [8]. In the push to leave no one behind with immunization services, there is a growing need to reach “zero-dose children”, those who have not received any routine vaccinations (measured by the lack of DTPcv1), as well as “under-immunized children” (defined as those missing the third dose of DTPcv (DTPcv3)) [1]. These children often have limited access to primary healthcare and social services, limited economic and educational opportunities, and limited political representation [9,10,11]. Zero-dose and under-immunized children are at an increased risk during disease outbreaks and also often lack access to other basic services. Providing these children with immunization services can connect them to other health services and the associated economic and social benefits [9]. Even in countries with high vaccination coverage, immunity gaps might occur among people in racial and ethnic minority groups, religious groups, urban settings, remote rural locations, migrant/nomadic communities, or low socioeconomic status [12]. Data to look at these zero-dose and under-immunized communities may not be routinely captured by the immunization program, or there may be challenges with the use of available data for programmatic decision-making, such as poor data accessibility and lack of timely analysis [2,3].

One approach that is being increasingly recognized as effective in improving data use and quality for decision-making in public health programs is data triangulation, or the synthesis of multiple datasets [13,14,15]. Data triangulation integrates data sources to identify data quality and immunization program performance gaps, including immunization-coverage inequities. The results can be used to guide programmatic action in communities that are often missed across the spectrum of essential health services [12]. In 2019, the Strategic Advisory Group of Experts (SAGE) Working Group on Immunization and Surveillance Data Quality and Use suggested that data triangulation should be the standard for public health analyses and that, even in the absence of perfect data, combining many pieces of weaker evidence through triangulation can form a strong basis for more informed decision-making [2,3]. As a result, the World Health Organization (WHO), UNICEF, and the U.S. Centers for Disease Control and Prevention (U.S. CDC) developed global guidance titled *Triangulation for Improved Decision-Making in Immunization Programmes* to describe a triangulation process that can be used by Expanded Programme on Immunization (EPI) and Vaccine Preventable Disease Surveillance (VPDS) programs to develop questions, identify data sources, and interpret different data together considering underlying context and limitations [12].

Many data sources are generated within and outside the EPI and VPDS programs (e.g., routine administrative data, VPD case data, coverage surveys, vaccine supply, serosurveys, and population estimates). However, there are challenges with the use of these separate data systems for programmatic decision-making, including poor data accessibility and lack of timely analysis, contributing to immunity gaps and VPD outbreaks in populations with limited access to immunization and basic healthcare services [2,12].

Since 2018, the U.S. CDC, in consultation with immunization experts from the WHO, has supported Ministries of Health (MoH) to build the capacity of the EPI and VPDS workforce in Bangladesh, Rwanda, and Nigeria to perform data triangulation for evidence-based decision-making. Here, we describe the methods, example triangulation indicators, and electronic information systems implemented by these three countries for routine data triangulation in RI and VPD Surveillance (VPDS) programs. We aimed to document how data triangulation activities have improved the availability of key indicators used for identifying DTPcv, polio, and measles-containing vaccine (MCV) immunization coverage inequities and examine how these triangulation processes may be used to improve immunization program performance, identify coverage gaps, and support the characterization of zero-doseor under-immunized children at all levels of the healthcare system in Bangladesh, Rwanda, and Nigeria.

## 2. Methods

### 2.1. Data Sources

A desk review of secondary data sources was conducted in each of the three countries. The data sources reviewed included country-specific VPDS and routine immunization (RI) data sources, electronic reporting systems, data triangulation analyses and dashboards, triangulated indicators, project reports, conference abstracts, and conference and national workshop presentations.

### 2.2. Data Review and Analysis

A manual thematic content analysis was performed using Microsoft Excel and Microsoft Word (Microsoft 365 MSO, Version 2308) on desk review materials, using themes identified by three reviewers a priori to inform the country-specific approaches to data triangulation activities. These themes examined the approach to data triangulation implementation, including stakeholder engagement and partnerships in each country, the mechanisms of implementation, data sources and processes, and approaches to building capacity for data triangulation. We also examined the results of country-specific data triangulation activities, including data triangulation immunity gap and immunization and VPDS program-performance indicators, data triangulation technologies and projects, visualization of indicators, and current use.

The results were summarized into four key areas based on the identified a priori themes: (1) project conceptualization (development) and partnerships, (2) approach to data triangulation processes and indicator prioritization, (3) data triangulation capacity building efforts, and (4) successes and demonstrated potential for impact. Our findings for each country and the lessons learned from these four key areas are presented below.

## 3. Results

### 3.1. Bangladesh

#### 3.1.1. Project Conceptualization and Partnerships

In 2019, the Bangladesh Directorate General of Health Services (DGHS), with funding and technical support from the U.S. CDC, served as the first pilot country to assist in the development of the global guidance for the publication *Triangulation for Improved Decision-Making in Immunization Programmes* [12]. At an initial triangulation concept-development workshop in March 2019, the DGHS prioritized the following topic areas as the biggest challenges in the country’s national and subnational immunization programs: (1) measles immunity gaps, (2) program performance, and (3) target-population estimates. The initial workshop resulted in project commitment and participation from the DGHS EPI, Surveillance, and Management Information System units; the Civil Registration and Vital Statistics Division; and the Bangladesh Bureau of Statistics. Additional key technical partners included the WHO and UNICEF Bangladesh offices.

#### 3.1.2. Approach to Triangulation Processes and Indicator Prioritization

During the national workshop, a data-mapping exercise of relevant data sources was completed prior to the development of a triangulation analysis plan to investigate the three identified programmatic areas: (1) measles immunity gaps, (2) program performance, and (3) target-population estimates. The data mapping included the name of the data source, at what administrative level the data are collected, in which information system the data are available, the reporting frequency, and at which administrative levels the data are used. Table 1 summarizes the data sources identified in Bangladesh. From the data inventory, key triangulation indicators for each of the three programmatic areas were prioritized based on the data sources and variables available, key questions of programmatic interest, and expected trends across datasets [12]. A list of example triangulation indicators to identify immunity gaps is summarized in Table 2. These indicators were ultimately incorporated into the triangulation global guidance. The global guidance documents describe each indicator in depth, including the data sources or elements required, potential analysis outputs, and sample visualizations from anonymized countries and publicly available references through collaboration with global subject-matter experts.

Once available data sources were identified and key triangulation indicators were prioritized within an agreed-upon data-analysis plan, datasets from various data sources (e.g., vaccination stock and supply, EPI administrative coverage, serosurveys, etc.) (Table 1) and information systems were exported into Microsoft Excel to allow for necessary data elements to be compiled into one standardized format for analysis. Additionally, the triangulation of data elements within datasets available on the same electronic information system, such as District Health Information System 2 (DHIS2), could be analyzed within the DHIS2 platform. Visualizations of the prioritized triangulation indicators were then presented to key national-level and subnational-level stakeholders for review and interpretation. While WHO Bangladesh and U.S. CDC conducted national- and district-level triangulation analyses between March and December 2019, national partners were instrumental in providing existing program guidance, policies, and access to data. Triangulation visualizations and corresponding interpretations were summarized according to programmatic areas (program performance, immunity gaps, and immunization-program target-population estimates) in a PowerPoint presentation, along with recommended programmatic action items, and presented to key stakeholders at a final dissemination workshop held in Dhaka in December 2019.

More recently, in 2022, Bangladesh assessed zero-dose (ZD) and under-immunized (UI) children by conducting data triangulation of existing data, like those listed in Table 1 [16]. The triangulation analyses of existing data sources conducted to identify communities with ZD and UI children are described in Table 2. Further description of the methodology used to identify missed ZD communities can be found in the report “Country Learning Hub for Immunization Equity in Bangladesh: Findings from Rapid Assessment Bangladesh” [17].

**Table 1 vaccines-12-00646-t001:** Data sources used to conduct triangulation analyses in Bangladesh. Summary of data sources identified in Bangladesh to assess immunization program performance, identify immunity gaps, and assess immunization-program target-population estimates.

Data Source	Dataset Types
Global-population estimates	• World Population Projection (UNDP) [18]
Immunization-program target-population estimates based on census	• M.G.S. Uddin 2014 census projection (commissioned for National EPI program) • G. Feeney 2017 revised census projection (commissioned for National EPI program)
Immunization-program target-population estimates based on microplan	• EPI Annual Microplans (2012–2018)
Civil Registration and Vital Statistics	• Birth and Death Registration Information System (BDRIS), including Civil Registration and Vital Statistics (CRVS)
Bureau of Statistics Data	• BBS Census Projections (2011–2061) [19] • Sample Registration and Vital Statistics Surveys (annual)
Vaccine Stock and Supply	• Stock/supply (vials used, received, and available) • Wastage
Program Management	• Vaccination sessions held (variable in DHIS2) • Human resources • Stockouts
Vaccination Coverage	• EPI administrative coverage • WHO UNICEF Estimates of National Immunization Coverage (WUENIC) [20] • Coverage surveys, e.g., Demographic Health Surveys, Multiple Indicator Cluster Surveys (MICS), or others • SIA administrative coverage and post-campaign surveys
Surveillance	• Case-based and laboratory databases • Aggregate (passive) reporting systems, e.g., DHIS2 • Disease incidence reported to the Joint Reporting Form (JRF) [21]
Contextual Information	• Vaccination schedule and history of any changes (e.g., vaccine intro) • Major geo-political events (e.g., insecurity, mass migrations, and disasters)
Other data on population immunity or disease burden	• Serosurveys • Modeled estimates of coverage, population immunity, or disease burden

**Table 2 vaccines-12-00646-t002:** Triangulation analyses conducted by Bangladesh to identify communities with ZD and UI children. List of Bangladesh’s priority indicators used to identify immunity gaps in 2018 and ZD/UI children in 2022 [12,17].

Analyses Performed to Identify Immunity Gaps	Analyses Performed to Assess Zero-Dose and Under-Immunized Children
• Comparison of administrative coverage, WUENIC, and coverage surveys • Trends in vaccination coverage or immunity by age group/birth cohort • Geographic trends in vaccination coverage across different data sources • Surveillance performance and reported cases/outbreaks at the subnational level • Comparison of vaccination coverage and surveillance data • Measles epidemiology (age and vaccination status of cases) • Immunity gaps in special populations • Outbreaks, vaccine stock, and other contextual information • Vaccination coverage surveys • Serosurveys • Modeling studies	• Ranking of percent ZD (%) for top 10 districts and top 5 urban city corporations (CC) for 2014, 2015, 2016, and 2019 coverage surveys (calculated using DTPcv1 coverage) • Ranking of percent ZD (%) using DHIS2 EPI administrative data from 2019 to 2022 to identify the top 10 health facilities • Comparison of DHIS2 EPI administrative data with monthly EPI reports and microplan target-population estimates of top 10 health facilities with the highest percent ZD • Lot quality assurance sampling (LQAS) to confirm the clusters with a high percentage (>10%) of ZD or UI children according to survey data and DHIS2 data analyses

#### 3.1.3. Approach to Capacity Building

To build triangulation capacity in the country, a final workshop was conducted in December 2019 to present the triangulation findings, provide recommendations based on the findings, and train national and subnational immunization program staff on triangulation methodology. An additional training workshop was conducted for district-level WHO consultants serving as Surveillance and Immunization Medical Officers (SIMOs) to build capacity for frequent and potentially automated analyses of key triangulation indicators related to identifying immunity gaps (see Table 2). The SIMO training included a hands-on workshop that allowed SIMOs to conduct triangulation analyses within DHIS2 by using available data sources within their assigned districts, including immunization coverage and VPD Surveillance. SIMOs were able to disaggregate triangulation indicators by health facility, ultimately identifying data quality issues and potential immunity gaps requiring follow-up in the field. The Bangladesh exercise was paramount in developing and finalizing the global triangulation guidance, which has since been published [12].

#### 3.1.4. Successes and Demonstrated Potential Impact

WHO Bangladesh and the Ministry of Health used recommendations from the final triangulation workshop in 2019 to inform Bangladesh’s National Data Improvement Plan required by Gavi, the Vaccine Alliance (Gavi). The following activities were incorporated into their 2019 plan: data triangulation-capacity building for SIMOs, revision of the 2020 microplan guidance for establishing district- and health facility-level target-population estimates, and development of DHIS2 triangulation dashboards to enhance supportive supervision [16].

As a follow-up to the triangulation analyses used to identify communities with the highest number of zero-dose and under-immunized children in 2022, rapid community assessments were conducted to better understand the reason why vaccine doses were not given or missed. A rapid community assessment (RCA) is a process for quickly triangulating existing data to identify missed communities with ZD and UI children. The RCAs also collected vaccination demand-and-supply barrier data, identifiable challenges of ZD and UI children in these communities, and stakeholders’ suggestions to reduce ZD and UI children. Targeted interventions were then developed as part of the RCAs to reach these children (Figure 1) [17].

The RCAs were conducted between December 2022 and May 2023 in the missed communities identified through the triangulation of the Coverage Evaluation Survey (CES) and DHIS2 administrative immunization-coverage data. The RCAs identified five rural districts and one urban (city corporation) with zero-dose, under-immunized, and missed communities by utilizing the triangulation indicators listed in Table 2. Findings from the missed communities with high zero-dose and under-immunized children confirmed that the initially identified areas were mainly inhabited by those who lacked access to educational institutions and health centers. The transportation system and household condition of these areas were also inadequate. The most common profession for the head-of-households in these clusters was farmer, followed by service professional, except for the urban clusters where most residents were day laborers [17]. This information, which was identified through the data triangulation process and RCAs, can be used to inform more targeted interventions to reach these communities, such as Friday, evening, or holiday vaccination sessions to reach the children of working mothers or transportation-cost reimbursement for healthcare workers to conduct outreach.

### 3.2. Rwanda

#### 3.2.1. Project Conceptualization and Partnerships

In 2022, the Rwanda Biomedical Centre (RBC), with funding and technical support from the U.S. CDC, designed a live web-based DHIS2 RI and VPDS data triangulation dashboard. The triangulation dashboard aimed to streamline the integration of data from DHIS2 EPI and VPD Surveillance packages to assess program performance and investigate immunity gaps in Rwanda. The RBC provided leadership on the implementation of data triangulation processes and data and dashboard management, with backend development performed by the Health Information System Programme (HISP) of Rwanda, a global organization supporting DHIS2 implementation. Other technical partners included WHO Rwanda and the University of Oslo (UiO), with further support and coordination from the African Field Epidemiology Network (AFENET).

#### 3.2.2. Approach to Triangulation Processes and Indicator Prioritization

Dashboard customization occurred from November 2022 to July 2023 using an online, publicly available global triangulation dashboard protype designed by UiO in collaboration with the U.S. CDC and WHO [22]. Customization began with mapping the existing DHIS2 data packages for VPDS and RI, including aggregate Integrated Disease Surveillance and Response (IDSR), case-based (individual level) VPDS data, and EPI Electronic Immunization Registry (aggregate and individual).

RBC then adapted key VPDS and RI program indicators to be used for monitoring program performance and immunity gaps for measles, polio, and neonatal tetanus, following the established guidance in the WHO, UNICEF, and U.S. CDC publication, *Triangulation for Improved Decision-Making in Immunization Programmes* [12]. A workshop was held with key national-, provincial-, and district-level stakeholders to prioritize specific indicators that could be compared or analyzed together to provide greater insight into immunization program performance and immunity gaps, at what administrative level this data triangulation process should occur, and what the review processes should look like.

Custom scripts were designed to allow the exchange of data between VPDS and Electronic Immunization Registry data packages (Figure 2). Data elements from the different modules were used to create indicators and design programmatically informative dashboard visualizations. The dashboard underwent external review by global and national stakeholders during May–June 2023. The beta version of the dashboard was deployed at the national level for use by the surveillance and immunization programs in July 2023, with wider dissemination to thirteen districts in October 2023.

A total of 14 immunity-gap and 11 program-performance indicators were ultimately included in the dashboard. Dashboard indicators were designed to permit the disaggregation of data across healthcare administrative levels (e.g., national-level data could be further disaggregated to the district and health-facility levels) and by time (e.g., yearly versus monthly) to tailor indicators to different user levels and data analysis needs.

Key triangulation indicators for the identification of immunity gaps included the vaccination status of VPD cases (including measles and rubella, acute flaccid paralysis (AFP; poliomyelitis), and neonatal tetanus) by age and vaccine eligibility, vaccination dropout rates (e.g., the proportion of infants who begin the vaccination schedule but fail to complete it) for MR, polio and DTP vaccinations, and the number of zero-dose and under-immunized children by administrative level each month.

For immunization program performance, prioritized indicators included the numbers of measles and rubella, AFP, and neonatal tetanus cases reported in the aggregate IDSR system compared to the individual-level, case-based surveillance reporting system. Combining multiple data sources with different aggregation levels in one visualization can facilitate easier identification of data quality issues and issues with reporting. Additional indicators included the access to and utilization of immunization services by district, monthly. Figure 3 shows one of these program-performance visualizations: DTP1 coverage (signifying immunization service access issues) versus DTP1-DTP3 dropout rate (a high rate is indicative of many individuals not utilizing the immunization services offered) by district in Rwanda.

#### 3.2.3. Approach to Capacity Building

National and international VPDS, immunization, and Health Management Information System (HMIS) experts conducted a series of three-to-five-day data triangulation workshops between July and September 2023, at both national and district levels to promote a clear understanding and effective use of the DHIS2 triangulation dashboard. The training curriculum consisted of didactic materials, including presentations on the basics of data triangulation and triangulation for identifying immunity gaps and program performance challenges, as well as case studies and live demonstrations of the dashboard. Thirty-eight HMIS data managers, IDSR focal persons, and EPI supervisors from 14 of Rwanda’s 30 districts participated in these initial workshops (at the time of the training, only these 14 districts were utilizing the DHIS2 case-based module for VPDS). Real-time anonymous feedback was collected during workshops to assess the comprehension of the materials and the usefulness of the dashboard, as well as offer suggestions for improvement and use. In April 2024, mentorship sessions were conducted in the 14 trained districts. Mentors held discussions with district staff, and questionnaires were used to gather feedback regarding their usage frequency and experience with using the dashboard, and examine how the dashboard had assisted in addressing programmatic queries. According to the questionnaire, 43% of surveyed staff reported actively using the dashboard, with 21% reporting weekly use. However, the remainder of those surveyed reported that they were not actively utilizing the dashboard, primarily attributing this to a general lack of understanding of how to interpret dashboard indicators. Based on the user feedback received, a triangulation dashboard self-study guide was created to guide users through dashboard indicators, clarify meanings in their specific context, identify and record data quality issues and immunity gaps, and propose and track corrective actions.

#### 3.2.4. Successes and Demonstrated Potential for Impact

Multiple efforts have been made to improve our understanding and utilization of the DHIS2 data triangulation dashboard through national and subnational workshops and feedback-gathering sessions. One achievement has been the establishment of the Rwanda Immunization Data Technical Working Group (TWG). The TWG provides a forum for HMIS, immunization, and VPD Surveillance representatives from the national, provincial, and district levels to discuss current activities and issues around immunization-related data collection, entry, and analysis and the use of the DHIS2 immunization data triangulation dashboard. Examples of observations made by the TWG using the triangulation dashboard include the identification of the immunization status of measles cases by age group and geographic location during a measles outbreak, indicating which groups to target for vaccination, and multiple data quality discrepancies between case-based and aggregate VPD Surveillance reporting systems for measles. Additional plans by the TWG to ensure the use of the dashboard at the subnational level include incorporating dashboard reviews into supportive supervision and mentorship visits and monthly meetings between Provincial Health Emergency Operation Centers and District hospitals.

### 3.3. Nigeria

#### 3.3.1. Project Conceptualization and Partnerships

Nigeria’s National Primary Health Care Development Agency (NPHCDA) manages the EPI program, and RI data are reported through DHIS2. The national DHIS2 platform is hosted by the Federal Ministry of Health (FMOH). Comparably, the Surveillance Outbreak Response Management and Analysis System (SORMAS) platform, managed by the National Centre for Disease Control (NCDC), is used to report VPDS data. The reporting of immunization and surveillance data through multiple siloed systems and to different responsible agencies has contributed to a lack of data access, sharing, coordination, and use in Nigeria. In 2019, recognizing the need for improved data integration and use, NPHCDA, NCDC, and AFENET staff developed a data triangulation dashboard leveraging the U.S. CDC Growing Expertise in E-Health Knowledge and Skills (GEEKS) traineeship program [24]. The primary objective of the dashboard was to collate, analyze, and visualize RI and VPD Surveillance data from the DHIS2 and SORMAS platforms. Additional technical partners included the HISP of Nigeria, and NIX Technologies—an Information and Communications Technology (ICT) organization that supports NCDC in managing the SORMAS system in Nigeria.

#### 3.3.2. Approach to Triangulation Processes and Indicator Prioritization

A data-mapping exercise was conducted to identify the existing data elements and indicators in the DHIS2 and SORMAS RI and VPDS platforms. During the data-mapping process, indicator calculations, the administrative levels of data collection and use, and the frequency of reporting were reviewed. Stakeholders also reviewed performance indicators extracted from the WHO, UNICEF, and U.S. CDC guidelines on *Triangulation for Improved Decision-Making in Immunization Programmes* [12]. Stakeholders from FMOH, NPHCDA, NCDC, and AFENET created a prioritized list of indicators, analyses, and visualizations to incorporate into the triangulation dashboard. Indicator prioritization was guided by key program issues and relevant questions identified by stakeholders. Based on the programmatic issues identified, immunization coverage reports from multiple survey data conducted since 2013 were also included in the dashboard configuration. The results of the indicator prioritization activity, including the data sources and visualizations initially selected for inclusion in the triangulation dashboard, are shown in Table 3.

The dashboard was configured for four VPDs (measles, yellow fever, meningitis, and diphtheria). The selected indicators were visualized on a web-based dashboard developed using R, R Shiny, Python, and JavaScript (see Figure 4). The dashboard was integrated into the national SORMAS platform and stored in a cloud-based Structured Query Language (SQL) database.

**Figure 4 vaccines-12-00646-f004:**
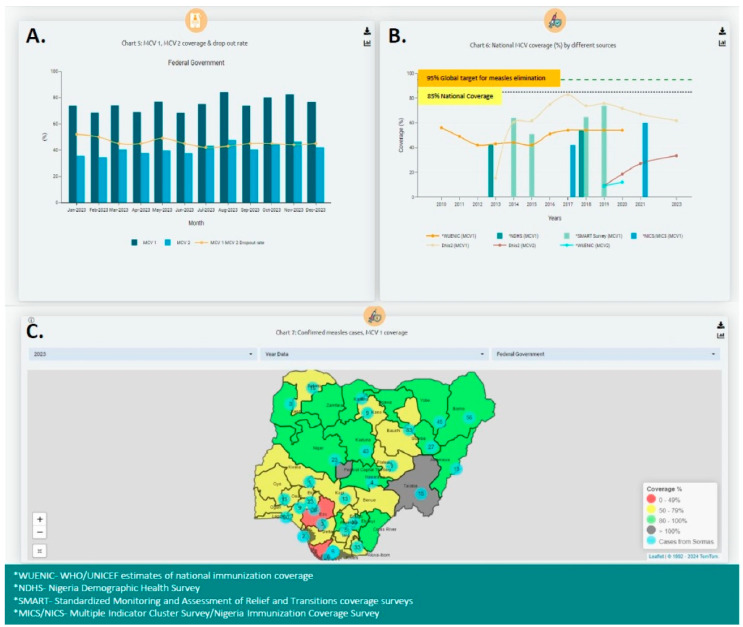
Screenshot displaying Nigeria’s R Shiny dashboard for DHIS2 RI, VPD Surveillance data from SORMAS, and vaccination coverage survey data. (**A**) Measles-containing vaccine first-dose (MCV1) and second-dose (MCV2) coverage and dropout rate. (**B**) MCV1 and MCV2 coverage by different data sources between 2010 and 2023 (administrative coverage, Nigeria Demographic Health Survey (NDHS), Multiple Indicator Cluster Survey/Nigeria Immunization Coverage Survey (MICS/NICS), Standardized Monitoring and Assessment of Relief and Transitions (SMART) coverage surveys, and WHO/UNICEF estimates of national immunization coverage (WUENIC)). (**C**) Maps of measles cases verses MCV1 coverage by state.

#### 3.3.3. Approach to Capacity Building

The RI/VPDS triangulation dashboard was designed and implemented through the GEEKS Nigeria traineeship program [19]. A one-week onboarding training was provided to all GEEKS fellows, followed by a 12-month dashboard design, customization, and implementation period. Regular mentorship sessions were held bi-weekly between mentors and fellows to build the capacity for data triangulation and on the use of DHIS2, SORMAS, R, Microsoft Suites, and additional relevant areas of data and project management.

#### 3.3.4. Successes and Demonstrated Potential for Impact

Several initiatives have been undertaken to ensure the ongoing use of the RI and VPDS dashboard at both the national and subnational levels in Nigeria. The dashboard has been integrated into SORMAS, and efforts are currently ongoing to embed it within the national DHIS2 platform, meaning that RI and surveillance staff at the national and subnational levels are able to access the dashboard and make use of the data for decision-making. A national TWG has also been established for NPHCDA and NCDC staff to review the dashboard monthly. Additionally, the dashboard has been implemented in two states in Nigeria (Yobe and Jigawa), where it is reviewed by subnational RI and surveillance staff during monthly TWGs to guide programmatic decision-making. Several actions taken by the national TWG based on these triangulated data have included the provision of feedback to state and local government areas (LGAs) on MCV-coverage data quality issues, the identification of geographical areas with measles outbreaks to support more targeted outbreak response and vaccination activities, and the identification of LGAs in need of supportive supervision.

## 4. Discussion

If the IA2030 targets to reduce the number of zero-dose children are to be met, immunization programs need to be able to accurately identify those children and take appropriate action [1,4,9,11,25]. In the context of immunization programs, greater data use can result in better quality data and ultimately contribute to improved program performance by better identifying and targeting those who are eligible for vaccination [2,12,26,27].

Here, we described how three countries have taken a data-driven approach to identify immunization coverage inequities and program performance challenges by using triangulation analyses and dashboards for identifying DTPcv, polio, and MCV immunization coverage gaps at all levels of the healthcare system. Even in the absence of perfect data, combining many parts of less robust evidence through triangulation can provide a strong basis for more informed decision-making.

## 5. Lessons Learned and Recommendations

To further strengthen the use of data triangulation for improving health equity in immunization and VPDS programs, we provide several lessons learned and recommendations based on findings from our four a priori themes.

### 5.1. Early Engagement and Ongoing Coordination across Programs and Stakeholders at Each Level Is Critical

Since the success of triangulation is contingent on access to and use of many data sources, a high degree of cooperation and buy-in are required from multiple institutions and stakeholders. Triangulation is most effective when stakeholders are involved at all stages, including prioritizing the questions to be answered, identifying the data sources, guiding the analysis and interpretation, and using the results for decision-making in their policies and programs. Collective engagement from national and subnational immunization, VPDS, and HMIS staff early in the development process will help to promote the cross-program prioritization of triangulation indicators and integration into existing program activities to ensure ongoing use and prevent any additional burden on health staff.

### 5.2. Establish Regular Processes for Reviewing and Using Triangulated Data

Having regular processes and accountability mechanisms in place that support data access, quality, interoperability, and use is important for establishing useful triangulation analyses. In all three countries, establishing triangulation processes with imperfect data allowed stakeholders to work collaboratively and increase their comfort and capacity to use triangulated data to develop targeted interventions and corrective actions. Creating a strong “data-use culture” from the local to the national level can result in better-quality data and ultimately contribute to improved immunization program performance.

### 5.3. Continued Capacity Building for Triangulation Analysis and Use for Action Are Needed at All Levels, Even after Electronic Tools and Processes Are Established

Dashboards and tools cannot perform all the triangulation analyses and interpretation for the end-user. Data use and critical thinking are required for staff to synthesize and contextualize data into actionable recommendations. Ongoing capacity building for triangulation analysis and data use beyond trainings are required, such as incorporating dashboard reviews into supportive supervision visits, regular review meetings at national and subnational levels, and other accountability mechanisms. These will help to equip both national and subnational staff with the skills they need to make decisions and take actions based on analyses and interpretations to vaccinate under-immunized communities and reach zero-dose children.

### 5.4. The Greatest Successes and Impact Occur When There Is Collaboration and Utilization of Triangulated Data by Health Staff and Policymakers across Programs and Healthcare Levels

While Bangladesh, Nigeria, and Rwanda were able to conduct data triangulation analyses, develop dashboards, and integrate multiple data sources, the greatest successes were due to the collaboration and utilization of information by health staff and policymakers across programs and healthcare levels. In Rwanda and Nigeria, immunization and VPDS data-use working groups were established to further promote the use of data across stakeholders and programs in order to best address and support targeted vaccination-response activities and areas in need of supportive supervision. In Bangladesh, triangulation was used to identify missed communities with high zero-dose and under-immunized children, which led to targeted rapid convenience assessments to identify barriers to reaching them so that effective and targeted interventions could be developed. To further optimize the impact, triangulation analyses and interpretation could be included in existing regular periodic activities where analysis is already performed, such as monthly data-review meetings, annual desk reviews, National Immunization Technical Advisory Group (NITAG) meetings, or National Committees for certifying polio eradication or verifying measles and rubella elimination.

## 6. Considerations and Limitations

This manuscript describes data triangulation processes in only three countries. The results are not generalizable to all settings and unique country challenges. Apart from the limited capacity for conducting and interpreting triangulation analyses, many reasons may exist for not effectively using data, from data-access challenges to the lack of a supportive “data-use culture”. Triangulation is not a singular solution to these larger problems. However, actions increasing data access, use, and understanding may lead to gradual improvements over time.

Additionally, it is important to consider local knowledge and contextual information to further interpret the available data, including explanatory causes, and develop targeted program-improvement efforts. Unexpected or contradictory findings may emerge. However, local and contextual knowledge can provide additional insights into the reasons behind data discrepancies or conflicting evidence, leading to a more nuanced understanding of the programmatic question.

Lastly, since no formal assessments or evaluations of RI and VPDS triangulation activities have been conducted in these three countries since triangulation tools and processes have been established, it is not yet possible to measure the extent to which their implementation has directly impacted data use and quality, or how triangulated data have been used for programmatic action. Such evaluations to measure data triangulation outcomes and document impact more formally are needed.

## 7. Future Directions and Needs

Additional initiatives to support WHO Regional Offices and countries at both the national and subnational levels to incorporate triangulation sessions and experience-sharing into existing data workshops and trainings are needed. These should involve stakeholders across multiple programs where possible (e.g., immunization, surveillance, HMIS, national statistics offices, birth/civil registration offices, and other relevant organizations) which would help to promote immunization program coordination. It is critical that any new data triangulation initiatives for identifying zero-dose and under-immunized children be integrated with larger, multi-sectoral efforts to improve overall immunization performance.

Likewise, an increasing number of low- and middle-income countries are leveraging the availability of new electronic information technologies for immunization and VPDS data management, including electronic immunization registries (EIRs), electronic Logistics Management Information Systems (eLMISs), and the DHIS2 VPD CBS package, which have the potential to improve the quality, timeliness, and use of data [2,28,29,30]. However, these new electronic systems are not magic solutions to more systemic problems, and they are unlikely to lead to lasting programmatic improvements unless other factors are considered, such as infrastructure, strong national governance and coordination, and workforce capacity [29,30]. Additionally, the integration and interoperability of newly introduced systems are crucial to ensuring that all available data can be leveraged. By enhancing integration and interoperability, the exchange, access, and utilization of data can be significantly improved. Additional efforts to support the interoperability between data systems for easier triangulation analyses and integration of these systems, such as VPD CBS with laboratory information systems, will be critical. At the country-level, planning across programs to ensure the integration and interoperability of any newly introduced tools within the existing information system should also be prioritized.

## 8. Conclusions

In order to address inequities in immunization coverage and protect zero-dose and under-immunized children against VPDs, sustainable improvements in data quality and use are needed at all levels. Here, we described how data triangulation was used by three countries to identify immunity gaps, detect zero-dose and under-immunized children, and assess program performance at all levels of the healthcare system. Triangulation methodologies and processes, as well as lessons learned, may be leveraged by different country contexts and incorporated into routine RI and VPDS data analyses, potentially leading to programmatic changes that promote improved access to vaccination services.

## Figures and Tables

**Figure 1 vaccines-12-00646-f001:**
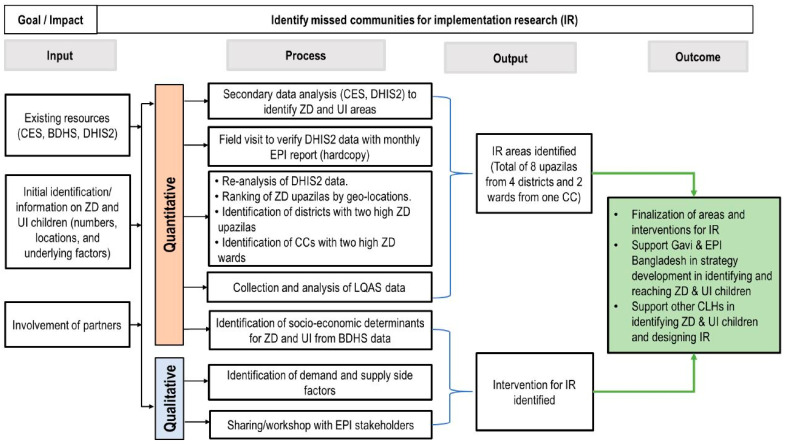
Conceptual framework for the rapid assessment of zero-dose (ZD) and under-immunized (UI) children. The RCA conceptual framework is used for quickly triangulating existing data to identify missed communities and develop more targeted interventions to reach ZD and UI children. Triangulation is an integral part of the quantitative methodology used to identify areas in which to conduct RCAs, with qualitative information also used to develop strategies for identifying and reaching ZD and UI children [17]. Acronyms: BDHS, Bangladesh Demographic and Health Survey; CC, city corporation; CES, Coverage Evaluation Survey; EPI, Expanded Programme on Immunization; IR, implementation research; CLH, Country Learning Hub.

**Figure 2 vaccines-12-00646-f002:**
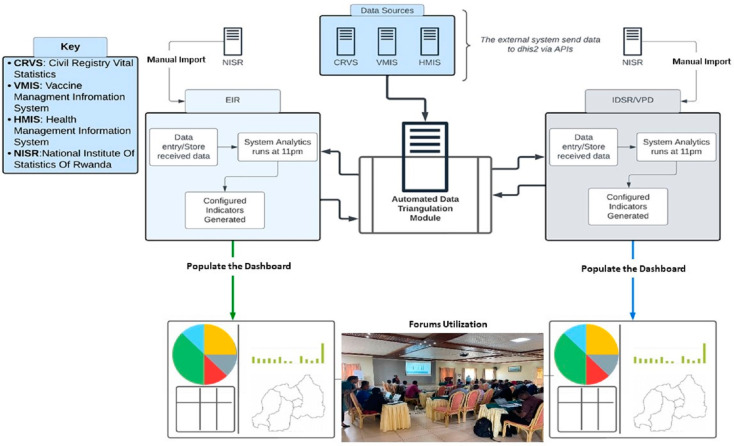
Approach to the triangulation of RI and VPDS data sources in Rwanda. Diagram showing the approach to the triangulation of routine immunization (RI) Electronic Immunization Registry (EIR), and Integrated Disease Surveillance and Response (IDSR)/case-based vaccine preventable disease surveillance (VPDS) data sources within Rwanda’s health management information system (HMIS) and external data sources utilized by Rwanda. Integration of the Laboratory Information System (LIS), electronic Civil Registration and Vital Statistics (CRVS), and Vaccine Logistics Management Information System (vLMIS) datasets into the triangulation dashboard is ongoing [23].

**Figure 3 vaccines-12-00646-f003:**
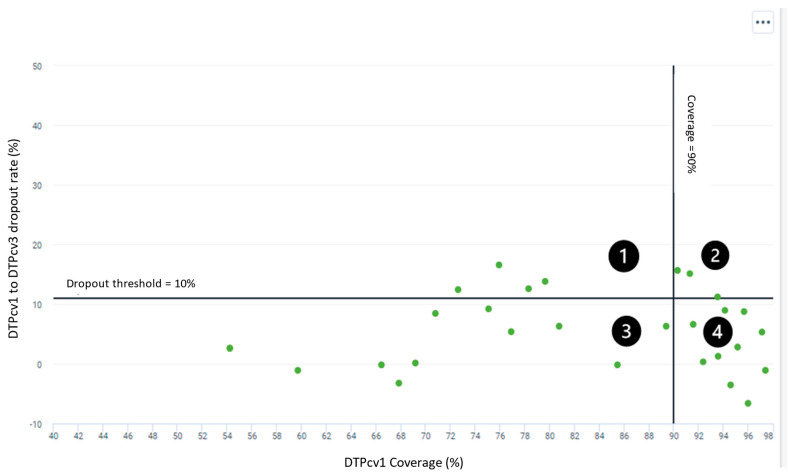
Scatter plot of monthly DTPcv1 coverage (%) versus DTPcv1-DTPcv3 dropout rate by district in Rwanda. (**1**) Low DTPcv1 coverage (<90%) and high dropout (>10%) = access and utilization issues. (**2**) High DTPcv1 coverage but high dropout = utilization issues. (**3**) Low DTPcv1 coverage but low dropout = access issues. (**4**) High DTPcv1 coverage and low dropout = No access and utilization issues. Each green dot is representative of a district in Rwanda.

**Table 3 vaccines-12-00646-t003:** Triangulation analyses conducted to identify program performance and immunity gaps in Nigeria. Prioritized indicators, data sources, and visualizations included in Nigeria’s RI and VPDS data triangulation dashboard.

Indicator	Visualization	Disease	Data Source
Confirmed cases versus admin coverage	Combo chart	Measles/yellow fever/ meningitis	DHIS2 and SORMAS
Age group of confirmed cases by vaccination status	Stacked column chart	Measles/yellow fever/ meningitis	SORMAS
Vaccine stock analysis and admin coverage	Combo chart	Measles/yellow fever/ meningitis	DHIS2
Dropout rate	Combo chart	Measles	DHIS2
Discrepancy between co-administered vaccine doses	Combo chart	Measles/yellow fever/ meningitis	DHIS2
Confirmed versus admin coverage	Map	Measles/yellow fever/ meningitis	DHIS2 and SORMAS
National measles coverage by different data sources	Combo chart	Measles/yellow fever/ meningitis	DHIS2, SORMAS, WHO/UNICEF estimates of national immunization coverage (WUENIC), Standardized Monitoring and Assessment of Relief and Transitions (SMART), Nigeria Demographic and Health Survey (NDHS), Multiple Indicator Cluster Survey (MICS)/National Immunization Coverage Survey (NICS)

## Data Availability

No new data were created as part of this report. Data sharing is not applicable to this article.

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
