# Peer review of "Building Data Triangulation Capacity for Routine Immunization and Vaccine Preventable Disease Surveillance Programs to Identify Immunization Coverage Inequities"

_vaccines, 2024, doi:10.3390/vaccines12060646_

Round 1
Reviewer 1 Report
Comments and Suggestions for Authors
The manuscript " Building Data Triangulation Capacity for Routine Immunization and Vaccine Preventable Disease Surveillance Programs to Identify Immunization Coverage Inequities" reported the project/effort that the CDC led to improve the capability to use data triangulation process to determine the priority of indicators on vaccine-preventable diseases in Bangladesh, Nigeria, and Rwanda. It is a project with significant public health benefits. It is important to recognize the importance of this project.
Here are a few suggestions on method description and result presentation to present this important work effectively
1. Method part.
As this is a project report, it would be useful to describe how the funding, training, and mentorship programs were set up, launched, and evaluated. Therefore, readers will understand the project thoroughly.
2. Results part:
It would be helpful to view the result with program evaluation results in numbers, such as how many training was held, the number of trainees and mentorship pairs, the dashboard built, and a comparison of vaccine coverage before and after the project.
Tables need to provide more concise information. For example, table 1, "Bureau of Statistics Data," is where the data is from but not the dataset type. It would be helpful to see what dataset you can get from the listed data source. Please go over all the tables to make sure all the information listed is concise and easy to understand.
Overall, this project has significant public health value. However, the structure and content need to be adjusted to present effectively. It will need a major revision of the original paper on structure and content. I look forward to reading the revised version.
Comments on the Quality of English Language
Proofreading is needed for errors, such as extra space. and also writing with concise contents to make the manuscript easier to read.
Author Response
Reviewer 1
Dear Reviewer,
Thank you for your thoughtful review of our manuscript entitled “Building Data Triangulation Capacity for Routine Immunization and Vaccine Preventable Disease Surveillance Programs to Identify Immunization Coverage Inequities”, submitted to the Vaccines Special Issue “Inequality in Immunization 2024”. We appreciate the time and effort you have dedicated to providing this helpful feedback. Below, we address each of your comments and suggestions:
Comments and Suggestions for Authors
The manuscript " Building Data Triangulation Capacity for Routine Immunization and Vaccine Preventable Disease Surveillance Programs to Identify Immunization Coverage Inequities" reported the project/effort that the CDC led to improve the capability to use data triangulation process to determine the priority of indicators on vaccine-preventable diseases in Bangladesh, Nigeria, and Rwanda. It is a project with significant public health benefits. It is important to recognize the importance of this project.
Here are a few suggestions on method description and result presentation to present this important work effectively.
- Method part.
As this is a project report, it would be useful to describe how the funding, training, and mentorship programs were set up, launched, and evaluated. Therefore, readers will understand the project thoroughly.
Thank you for this comment. In our methods section, rather than describe the country-specific details regarding data triangulation project implementation (e.g., training, mentorship, etc.), we aimed to document how secondary data were collected, analyzed, and used to inform the results for each country (e.g., project conceptualization and partnerships, approach to data triangulation processes and indicator prioritization, data triangulation capacity building efforts, and successes and demonstrated potential for impact). Training, mentorship, and other country-specific information related to project implementation and demonstrated potential for impact are described in further detail in our results section for each country.
Additionally, at this time, no formal assessments or evaluations of triangulation activities have been conducted in these three countries since data triangulation dashboards and projects have been established, so we were not able to include this information in the manuscript. This information has been added to lines 518-523 “Since no formal assessments or evaluations of RI and VPDS triangulation activities have been conducted in these three countries since the triangulation tools and processes have been established, it is not yet possible to measure the extent to which their implementation has directly impacted data use and quality, or how triangulated data have been used for programmatic action. Such evaluations to measure data triangulation outcomes and document impact more formally are needed”.
Based on journal guidelines, we have also disclosed triangulation project funding information at the end of the manuscript. We have specified both country triangulation projects and manuscript funding sources in lines 569-570: “Funding: Country data triangulation projects and this work were funded by the U.S Centers for Disease Control and Prevention”.
2. Results part:
It would be helpful to view the result with program evaluation results in numbers, such as how many training was held, the number of trainees and mentorship pairs, the dashboard built, and a comparison of vaccine coverage before and after the project.
Thank you for this suggestion. We agree that evaluations of these data triangulation projects would be beneficial to assess the extent to which their implementation has directly impacted data use and quality, and how triangulated data have been used for programmatic action. To date, no formal assessments or evaluations of RI and VPDS triangulation activities have been conducted in these three countries since triangulation dashboards and processes have been established. Such evaluations to measure data triangulation outcomes and document impact more formally are needed. We have included this in our limitations section (lines 518-523) “Since no formal assessments or evaluations of RI and VPDS triangulation activities have been conducted in these three countries since triangulation tools and processes have been established, it is not yet possible to measure the extent to which their implementation has directly impacted data use and quality, or how triangulated data have been used for programmatic action. Such evaluations to measure data triangulation outcomes and document impact more formally are needed”.
However, some data related to the number of trainings, trainees, and results from the district level mentorship session were newly available for Rwanda. We have added this information to lines 332-349 “Thirty-eight Health Management Information System (HMIS) data managers, IDSR focal persons, and EPI supervisors from 14 of Rwanda’s 30 districts participated in these initial workshops (at the time of the training, only these 14 districts were utilizing the DHIS2 case-based module for VPDS). Real-time anonymous feedback was collected during workshops to assess comprehension of the materials, usefulness of the dashboard, and suggestions for improvement and use. In April 2024, mentorship sessions were conducted in the 14 trained districts. Mentors held discussions with district staff and questionnaires were used to gather feedback regarding their usage frequency, experience with using the dashboard, and examine how the dashboard had assisted in addressing programmatic queries. According to the questionnaire, 43% of surveyed staff reported actively using the dashboard, with 21% reporting weekly use. However, the remainder of those surveyed reported they were not actively utilizing the dashboard, primarily attributing this to a general lack of understanding of how to interpret dashboard indicators. Based on the user feedback received, a triangulation dashboard self-study guide was created to guide users through dashboard indicators, clarify meanings in their specific context, identify and record data quality issues and immunity gaps, and propose and track corrective actions”.
3. Tables need to provide more concise information. For example, table 1, "Bureau of Statistics Data," is where the data is from but not the dataset type. It would be helpful to see what dataset you can get from the listed data source. Please go over all the tables to make sure all the information listed is concise and easy to understand.
Thank you for this comment. We have reformatted the three tables so the presented information is clearer. For Table 1 “Bureau of Statistics Data”, the specific datasets are listed in the adjacent column (BBS Census Projections (2011-2061) and Sample Registration & Vital Statistics Surveys (annual)). To clarify this, we have renamed the two column titles to “Data Source” and “Dataset Types”.
Overall, this project has significant public health value. However, the structure and content need to be adjusted to present effectively. It will need a major revision of the original paper on structure and content. I look forward to reading the revised version.
We have added additional details as suggested and have addressed questions regarding the structure and clarity of our methods, results, and tables in the three comments above. We have also revised the formatting issues mentioned (such as proofreading for additional spaces, etc.).
At this time, programmatic evaluation results are not available and we were not able to include more robust impact data in the results or discussion sections. However, we have tried to clarify this as a significant limitation in our discussion.
Thank you once again for your valuable feedback. We hope the revisions address your concerns. We have also attached the revised tracked version of the manuscript for you to view.
Best regards,
Audrey Rachlin, PhD, MSc
Epidemiologist
Global Immunization Division
Centers for Disease Control and Prevention (CDC)

Reviewer 2 Report
Comments and Suggestions for Authors
I enjoyed reading this paper. For many public health problems - not just vaccination coverage - data triangulation provides the best assessment of the problem that is being considered.
I have a few minor points, however:
1. I suggest to include a statement in the Abstract on the fact that data triangulation requires a lot of ad hoc/ local/ problem specific knowledge and therefore can never be a cheap "routine" exercise.
2. I suggest the avoidance of "woke" words such as "equitable" (which carry some moral connotations) and replace them with more neutral terms, such as gaps in coverage.
3. Data triangulation is often presented as the bringing together of complementary data sets. It may happen, however, that data sources provide conflicting evidence. It might be a good idea to say something about how these conflicts can be resolved.
4. Similarly, it might be useful to say something (perhaps in the context of one of the three country studies) about bringing together data sources with different aggregation levels.
Author Response
Reviewer 2
Dear Reviewer,
Thank you for your thoughtful review of our manuscript entitled “Building Data Triangulation Capacity for Routine Immunization and Vaccine Preventable Disease Surveillance Programs to Identify Immunization Coverage Inequities”, submitted to the Vaccines Special Issue “Inequality in Immunization 2024”. We appreciate the time and effort you have dedicated to providing this helpful feedback. Below, we address each of your comments and suggestions:
Comments and Suggestions for Authors
I enjoyed reading this paper. For many public health problems - not just vaccination coverage - data triangulation provides the best assessment of the problem that is being considered.
I have a few minor points, however:
1. I suggest to include a statement in the Abstract on the fact that data triangulation requires a lot of ad hoc/ local/ problem specific knowledge and therefore can never be a cheap "routine" exercise.
Thank you for this suggestion. We agree that is important to include this as a consideration of data triangulation. We feel this is most appropriate to add to the “Considerations and Limitations” section. To lines 512-517 we have added “Additionally, it is important to consider local knowledge and contextual information to further interpret the available data, including explanatory causes, and develop targeted program improvement efforts.”.
Data triangulation does not need to be an expensive or time-consuming activity, particularly if the analysis builds on previous work. If preparing triangulation analyses for the first time, there may be an initial time or cost investment to prepare the data sources and analysis template. Once an initial analysis has been done, updating can be quick, easy, and yield good results. Analyses could be further automated for routine use.
2. I suggest the avoidance of "woke" words such as "equitable" (which carry some moral connotations) and replace them with more neutral terms, such as gaps in coverage.
Thank you for this suggestion. We have replaced both instances of equitable in the manuscript. In lines 35-37 of the Abstract section, we have changed the wording to “Triangulation processes and lessons learned may be leveraged by other countries, potentially leading to programmatic changes that promote improved access and utilization of vaccination services through the identification of UI and ZD children”. In lines 555-559 of the Conclusion, we have changed this to “Triangulation methodologies and processes, as well as lessons learned, may be leveraged by different country contexts and incorporated into routine RI and VPDS data analyses, potentially leading to programmatic changes that promote improved access to vaccination services.”
3. Data triangulation is often presented as the bringing together of complementary data sets. It may happen, however, that data sources provide conflicting evidence. It might be a good idea to say something about how these conflicts can be resolved.
Thank you for this suggestion. We agree that is important to include this as a consideration of data triangulation. To the “Considerations and Limitations” section (lines 514-517) we have added “Unexpected or contradictory findings may emerge. However, local and contextual knowledge can provide additional insights into the reasons behind data discrepancies or conflicting evidence, leading to a more nuanced understanding of the programmatic question.”
4. Similarly, it might be useful to say something (perhaps in the context of one of the three country studies) about bringing together data sources with different aggregation levels.
We agree that it would be useful to describe this in further detail in the manuscript. To the Rwanda “Approach to Triangulation Processes and Indicator Prioritization” section (lines 294-297) we have added “Dashboard indicators were designed to permit the disaggregation of data across healthcare administrative levels (e.g., national-level data could be further disaggregated to the district and health facility levels) and by time (e.g., yearly versus monthly) to tailor indicators to different user levels and data analysis needs.” To lines 305-310 we have also added “For immunization program performance, prioritized indicators included the numbers of measles and rubella, AFP, and neonatal tetanus cases reported in the aggregate IDSR system compared to the individual level case-based surveillance reporting system. Combining multiple data sources with different aggregation levels in one visualization can facilitate easier identification of data quality issues and issues with reporting. Additional indicators included…”
Thank you once again for your valuable feedback. We hope the revisions address your concerns. We have also attached the revised tracked version of the manuscript for you to view.
Best regards,
Audrey Rachlin, PhD, MSc
Epidemiologist
Global Immunization Division
Centers for Disease Control and Prevention (CDC)

Round 2
Reviewer 1 Report
Comments and Suggestions for Authors
While it is regret that the program evaluation data is not available, I understand that "since no formal assessments or evaluations of RI and VPDS triangulation activities have been conducted in these three countries since triangulation tools and processes have been established," I hope to see the program evaluation data in the future report! Good luck on future effort on this project!
Comments on the Quality of English LanguageGood.